DATA RELEASE

# The genome of the reef-building coral *Porites harrisoni* from the southern Persian/Arabian Gulf

Anna Fiesinger[1], Abdoallah Sharaf[2], Rachel Alderdice[1], Gabriela Perna[1], Hannah Manns[1], John A. Burt[3] and Christian R. Voolstra[1,*]

1 Department of Biology, University of Konstanz, Konstanz, Germany
2 SequAna Core Facility, Department of Biology, University of Konstanz, Konstanz, Germany
3 Center for Genomics and Systems Biology (CGSB) and Mubadala Arabian Center for Climate and Environmental Sciences (Mubadala ACCESS), New York University Abu Dhabi, Abu Dhabi, United Arab Emirates

## ABSTRACT

We present a genome assembly from the coral species *Porites harrisoni* from the southern Persian/Arabian Gulf, the hottest ocean basin where corals live. The assembly is 626.7 Mb in size, spanning 1,883 contigs with a contig N50 of 807.4 kb, including a single-contig mitochondrial genome. The assembly has a BUSCO completeness of 86.3% (single = 72.5%, duplicated = 13.7%, fragmented = 1.2%, missing = 12.5%). Within the nuclear genome, 59.23% are repeats (15.89% retroelements, 10.00% DNA transposons, and 31.71% unclassified repeats). Gene annotation of the nuclear genome assembly identified 27,823 protein-coding genes. The mitogenome is 18,639 bp long, with 13 protein-coding genes, 2 tRNAs, and 2 rRNAs. The *P. harrisoni* genome is a valuable resource of a coral from an extreme environment, enhancing understanding of the genomic architecture underlying thermal resilience. Comparative analyses will help elucidate the evolutionary basis of heat tolerance and the adaptive capacity of coral to rapid climate change.

**Subjects** Genetics and Genomics, Marine Biology, Evolutionary Biology

**Submitted:** 19 November 2025

* Corresponding author. Email: christian.voolstra@uni-konstanz.de

Preprint submitted at https://doi.org/10.64898/2026.02.26.708201

## INTRODUCTION

Corals surviving in extreme environments (i.e., experiencing variable and extreme water temperatures as well as high salinities) can serve as model organisms to study the adaptive capacity of corals in more temperate regions. Such corals offer valuable insights into the genomic architecture and genes underlying thermal resilience [1–3]. The hottest oceanic basin where corals occur is the shallow and semi-enclosed Persian/Arabian Gulf (PAG), where temperature extremes and ranges exceed those of coral reefs elsewhere (<12 °C to >36 °C annually) [4, 5]. Due to the relatively recent geological formation of the PAG, estimated at about 6–12 ka, and its shift to a hot climate, corals in these waters have had little time to adapt to these extreme conditions [6]. As a result, they inform our understanding of how coral reefs at large may respond to rapid warming.

One prominent coral in the PAG, and among the species most resilient to recurrent bleaching events [7], is the endemic species *Porites harrisoni* Veron, 2000 (Figure 1) from the complex coral clade. The complex clade comprises many ecologically dominant reef builders [8, 9], with the genus *Porites* hosting several thermally resilient species [10–14]. Colonies of *P. harrisoni* abundantly inhabit shallow fringing reefs in the PAG [15], and

**Figure 1. The species *Porites harrisoni*, Veron 2000.**
(A) Cladogram illustrating evolutionary relationships of *Porites harrisoni* (bold) and other Hexacorallia with sequenced genomes [28–35]. The data is adapted from a simple species tree generated with the NCBI ETE v3.1.3 toolkit [36]. Robust and complex coral clades are indicated [8, 9, 37]. A full-scale phylogenetic tree illustrating evolutionary relationships of the genus *Porites* within the Scleractinia is available from a recent study [38]. (B) *In situ* colony photo of *P. harrisoni* from Saadiyat Reef in the PAG. Picture credit: Christian R. Voolstra.

*P. harrisoni* exhibits diverse growth forms – submassive, nodular, columnar, or branching – on a broad encrusting base, with coloration varying from light to dark brown [15]. The thermal resilience of *P. harrisoni* is in part attributed to its association with the thermotolerant symbiotic alga *Cladocopium thermophilum*, especially in the southern PAG [6, 16, 17]. This alga has undergone rapid adaptive radiation upon colonizing the PAG [18]. In line with this, several studies suggest that PAG corals carry unique genomic signatures underpinning their exceptional heat tolerance [19, 20], conferring heritable thermal tolerance traits [20, 21]. Like many other corals in the PAG, *P. harrisoni* has one of the highest ecological bleaching thresholds in the world [22]. Originally described as a different species, *P. harrisoni* was first identified for its exceptional tolerance to extreme temperatures in a study highlighting the thermal tolerance of PAG corals [23] and later recognized for its persistence in hypersaline, thermally extreme lagoons where most corals are unable to survive [24]. However, this species also lives exceptionally close to its upper thermal tolerance threshold [25]. This species may provide critical insight into the genomic basis of coral thermal resilience, as it has persisted under harsh conditions that have eliminated many other corals, particularly in the southern PAG, due to recurrent bleaching events [12]. This understanding is crucial given the increasing rate of ocean warming.

Here, we report on the sequencing, assembly, and annotation of the genome of *P. harrisoni* from the southern PAG. Species taxonomy for this species is Eukaryota; Opisthokonta; Metazoa; Eumetazoa; Cnidaria; Anthozoa; Hexacorallia; Scleractinia; Fungiina; Poritidae; *Porites*; *Porites harrisoni* Veron, 2000 (NCBI:txid627007). The genome assembly is based on long-read Oxford Nanopore Technologies (ONT) sequencing. The generated genome is 626.7 Mb, comprising 1,883 contigs with a contig N50 of 807.4 kb. The nuclear genome annotation yielded 27,823 protein-coding genes, validated through RNA-Seq data and available curated Metazoa proteomes. We also assembled a complete single-contig mitochondrial genome (mitogenome), which should inform biodiversity assessment and further contribute to resilience understanding [26, 27]. The genome of *P. harrisoni* will facilitate comparisons among coral species regarding the genomic basis of thermal resilience.



## MATERIAL AND METHODS

### Sample collection, DNA isolation, and sequencing

One colony of *Porites harrisoni* was tagged and sampled for genomic DNA (gDNA) extraction for the reference genome (BioSample ID SAMN41390914; Table 1) at Al Saada Reef, Abu Dhabi, United Arab Emirates (24.085250, 52.243333) in the thermally extreme southern PAG on 30 May 2022 using a Nemo underwater drill (Nemo Power Tools LLC, Nevada, USA) fitted with a diamond tip corer. The coral plug was obtained from a Coral Bleaching Automated Stress System (CBASS) [39] experiment described elsewhere [25] at the baseline temperature (34.1 °C) treatment, at time point 2 (i.e., after overnight recovery). The coral fragment was submerged in 3 ml of DNA/RNA shield (Zymo Research Europe GmbH, Freiburg, Germany) in a WhirlPak bag for at least 3 hours, after which the tissue was airpicked off the skeleton using an air pump with sterile filter tips. Tissue slurry was transferred to 2 × 2 ml cryotubes and stored at 4 °C until shipment to the lab facilities at the University of Konstanz, Germany. The sample was stored at –20 °C until extraction.

DNA was extracted using the DNeasy Blood & Tissue kit (Qiagen, Hilden, Germany) following the manufacturer's protocol with a minor modification. Briefly, the tissue slurry was homogenized using a Polytron PT 1200 E (Kinematica, Switzerland); then, the sample was centrifuged at $5,000 \times g$ for 3 mins, and 90 µl of the supernatant were transferred from the cryotube to a 1.5 ml tube. Next, 90 µl of ATL buffer and 20 µl of Proteinase K (Qiagen, Hilden, Germany) were added, and the sample was lysed for 1 hr at 56 °C with shaking at 300 rpm. After extraction, two elution steps were performed [40]: 50 µl of AE elution buffer (Qiagen, Hilden, Germany) were added to the membrane of the spin column, incubated for 1 min, and centrifuged at $7,000 \times g$ for 1 min. A subsequent 50 µl AE buffer were added to the membrane of the spin column in the same elution tube, incubated for 5 mins, and centrifuged as previously. The quantity and quality of the DNA were assessed using a NanoDrop2000 spectrophotometer (Thermo Fisher Scientific, Waltham, Massachusetts, USA).

A total of 6.54 µg DNA were submitted to the NGS Competence Centre Tübingen (NCCT) for the construction of an ONT sequencing library using the ligation protocol LSK114 (Oxford Nanopore Technologies, Oxford, UK). The library was sequenced with an R10.4.1 flowcell on a PromethION 24 (Oxford Nanopore Technologies, Oxford, UK). Base calling was done using MinKNOW v23.07.12 (Oxford Nanopore Technologies, Oxford, UK) and Dorado v0.7.4. (RRID:SCR_025883) [41]. Base-called reads were trimmed using porechop v0.2.4 (RRID:SCR_016967) [42], and the quality was assessed using NanoPlot v1.44.1 (RRID:SCR_024128) [43]. For the genome assembly, the ONT reads were split into longer reads for assembly (minimum average quality 3, minimum length 1,000 bp) and shorter reads for polishing (minimum average quality 5, minimum length 500 bp), while removing reads mapping to algal symbionts (i.e., all publicly available reads from genomes of the dinoflagellate family Symbiodiniaceae) using chopper v0.9.0 (RRID:SCR_026486) [43]. The Symbiodiniaceae reference reads (NCBI:txid252141) were downloaded from NCBI using the NCBI Datasets command-line interface (CLI) v16.14.0 (parameters: genome taxon 252141) [44].

### Sample collection, RNA isolation, and sequencing

A total of 20 samples (~2 cm each) were drilled from 10 colonies of *Porites harrisoni* at Al Saada Reef (24.085250, 52.243333) in the southern PAG on 30 May 2022 for RNA isolation



(Table 1) using a Nemo underwater drill (Nemo Power Tools LLC, Nevada, USA) fitted with a diamond tip corer. The coral plugs were sampled from a CBASS experiment [39] described elsewhere [25], at the baseline temperature (34.1 °C) at time points 1 and 2, i.e., after the heat-hold and after the overnight recovery. As described above, the coral fragments were submerged in 3 ml of DNA/RNA shield (Zymo Research Europe GmbH, Freiburg, Germany) in a WhirlPak bag for at least 3 hours, after which the tissue was airpicked off the skeleton using an air pump with sterile filter tips. Tissue slurry was transferred to 2 ml cryotubes and stored at 4 °C until shipment to the lab facilities at the University of Konstanz, Germany. The samples were stored at –20 °C until extraction.

RNA was extracted following a previously described protocol [45] using the RNeasy Mini Kit (Qiagen, Hilden, Germany) adapted for use with a QIAcube (Qiagen, Hilden, Germany) for standardized extraction. The quantity and quality of the RNA were assessed using a NanoDrop2000 spectrophotometer (Thermo Fisher Scientific, Waltham, Massachusetts, USA) as well as gel electrophoresis to check for the presence and integrity of rRNA bands. The extracted RNA was sent to the NCCT for library preparation using the NEBNext Ultra II Directional mRNA kit and subsequent paired-end sequencing on an Illumina NovaSeq (2 × 150 bp).

## Estimation of the genome size
The genome size was estimated using a k-mer histogram generated with Meryl v1.3 (RRID:SCR_026366) [46] based on the ONT base-called and trimmed reads, and visualized in GenomeScope 2.0 (RRID:SCR_017014) [47].

## Genome assembly, curation, and evaluation
The initial draft genome was assembled using a subset of size-selected ONT reads (minimum average quality 3, minimum length 1,000 bp; total number of reads: 20,260,201) with the *de novo* Nanopore assembler NECAT v0.0.1 (RRID:SCR_025350) [48]. The quality of the assembled draft genome was assessed using BlobToolKit v4.3.0 (RRID:SCR_025882) [49], which prompted the removal of 442 contigs due to their similarity to non-related taxa. The draft assembly was then polished using a second subset of size-selected ONT reads (minimum average quality 5, minimum length 500 bp; total number of reads: 21,513,716) with one round of Racon v1.5.0 (RRID:SCR_017642) [50] and one round using Medaka v1.12.0 (parameters: –model r1041_e82_400bps_hac_v4.3.0) [51]. After polishing, funannotate v1.8.5 (RRID:SCR_023039) [52] was used to remove duplicated contigs and those shorter than 200 bp (parameters: funannotate clean, -m 200), prompting the removal of a further five contigs. The quality of the assembly was assessed using BUSCO v6.0.0 (RRID:SCR_015008) [53] and BlobToolKit v4.3.0.

## Mitochondrial genome assembly and annotation
The mitogenome was assembled using the same set of size-selected reads as above (minimum average quality 3, minimum length 1,000 bp), with a subsequent filtering step: reads were mapped to the available *Porites harrisoni* mitogenome (GenBank accession: MG754070, RefSeq accession: NC_037435) [54] using minimap2 v2.28 (RRID:SCR_018550) [55]. The mapped reads were then extracted using SAMtools v1.21 (RRID:SCR_002105) [56] and subsequently assembled using Canu v2.3 (RRID:SCR_015880) [57] (parameters: genomeSize=18k, -nanopore -trimmed). The resulting mitogenome was circularized using



Circlator v1.5.5 (RRID:SCR_016058) (parameters: –assembler canu, –merge_min_id 85, –merge_breaklen 1000) [58]. The circularized mitogenome was then polished using Racon v1.5.0 with the shorter reads as above (minimum average quality 5, minimum length 500 bp). The mitogenome was annotated with MITOS2 [59] and manually curated in Geneious Prime v2025.1.1 (RRID:SCR_010519) (GraphPad Software, LLC, Boston, USA) against a published *P. harrisoni* mitogenome [54] and a mitogenome of the closely related species *Porites lobata* [60].

### Repeat annotation
DNA repeats in the genome were identified using a combination of RepeatModeler v2.0.4 (RRID:SCR_015027) [61] and EDTA v2.2.2 (RRID:SCR_022063) [62] and masked using RepeatMasker v4.1.6 (RRID:SCR_012954) [63].

### Gene prediction and annotation
Gene prediction was performed using BRAKER3 v3.0.8 (RRID:SCR_018964) [64]. First, the transcript evidence was prepared as follows. After quality checking the RNA-Seq reads using FastQC v0.12.1 (RRID:SCR_014583) [65], the reads were trimmed using Trimmomatic v0.39 (RRID:SCR_011848) [66] and subsequently mapped to the *Porites harrisoni* assembly using STAR v2.7.11b (RRID:SCR_004463) [67]. The resulting mapping files were merged, and the reverse strand-specific RNA reads were extracted using SAMtools v1.21. Second, curated Metazoa proteomes were retrieved from OrthoDB 11 [68] and used as protein evidence as recommended by the BRAKER3 pipeline [64, 69, 70]. Lastly, BRAKER3 was run using the reverse strand-specific RNA mapping files and protein evidence as input with the masked genome assembly (parameters: –busco_lineage=metazoa_odb10, –species=Porites_harrisoni) [53, 71–80]. Using the script stringtie2utr.py from the BRAKER3 suite, untranslated regions (UTRs) were added to the annotation file output by BRAKER3. In addition, 674 eukaryotic high-confidence transfer RNAs were predicted using tRNAscan-SE v2.0.12 (RRID:SCR_008637) [81] based on filtering of the initial set of 5,938 putative tRNAs using EukHighConfidenceFilter [81]. The high-confidence set of tRNAs was merged with the BRAKER3 structural predictions. Then, the annotation file was checked for overlapping genes using AGAT (RRID:SCR_027223) [82] and validated using GenomeTools v1.6.5 (RRID:SCR_016120) [83]. Finally, GffRead (RRID:SCR_018965) [77] was used to extract the predicted protein sequences from the merged file for functional annotation. The predicted genes were annotated using InterProScan v5.59-91.0 (RRID:SCR_005829) [84], EggNOG-mapper v2.1.12 (RRID:SCR_021165) [85, 86], and Phobius v1.01 (RRID:SCR_015643) [87]. The respective annotation files were then fed into funannotate, with the predicted genes in the GFF3 file format, to synthesize all annotations (parameters: –species "Porites harrisoni" –busco_db metazoa, –eggnog –iprscan –phobius). The final annotation was assessed using BUSCO v6.0.0. The gene prediction and functional annotation outlined in this study were used to develop the fully automated annotation pipeline GeneForge v1.0 [88].

## RESULTS AND DISCUSSION
### Genome assembly
The genome was sequenced from a single *Porites harrisoni* colony, collected from Al Saada Reef in the southern PAG on 30 May 2022 (Table 1). A total of 22,825,985 ONT long-reads were obtained from an R10.4.1 flow cell (Oxford Nanopore Technologies, Oxford, UK)

**Table 1.** Specimen and sequencing data for *Porites harrisoni.*

| Project information | | | | |
|---|---|---|---|---|
| Study title | *Porites harrisoni* southern Persian/Arabian Gulf | | | |
| ToLID | jaPorHarr | | | |
| BioProject ID | PRJNA1111311 | | | |
| BioProject title | *Porites harrisoni* genome sequencing and assembly - DFG ATLAS | | | |
| Species | *Porites harrisoni* | | | |
| BioSample ID | SAMN41390914 | | | |
| NCBI taxonomy ID | 627007 | | | |
| **Specimen information** | | | | |
| Sample | Library type | Sequencing platform | BioSample ID | SRA accession number(s) |
| *Genome* | | | | |
| UAE_SA_Phar_03 | ONT library (ligation protocol LSK114) | ONT R10.4.1 | SAMN41390914 | SRR29049442 |
| *RNA-Seq* | | | | |
| UAE_SA_Phar_01 | NEBNext Ultra II Directional mRNA | Illumina NovaSeq | SAMN45077167 | SRR35924083 SRR35924084 |
| UAE_SA_Phar_02 | NEBNext Ultra II Directional mRNA | Illumina NovaSeq | SAMN45077168 | SRR35924075 SRR35924076 |
| UAE_SA_Phar_03 | NEBNext Ultra II Directional mRNA | Illumina NovaSeq | SAMN45077169 | SRR35924345 SRR35924346 |
| UAE_SA_Phar_04 | NEBNext Ultra II Directional mRNA | Illumina NovaSeq | SAMN45077170 | SRR35924573 SRR35924574 |
| UAE_SA_Phar_05 | NEBNext Ultra II Directional mRNA | Illumina NovaSeq | SAMN45077171 | SRR35924564 SRR35924565 |
| UAE_SA_Phar_06 | NEBNext Ultra II Directional mRNA | Illumina NovaSeq | SAMN45077172 | SRR35924455 SRR35924456 |
| UAE_SA_Phar_07 | NEBNext Ultra II Directional mRNA | Illumina NovaSeq | SAMN45077173 | SRR35924447 SRR35924448 |
| UAE_SA_Phar_08 | NEBNext Ultra II Directional mRNA | Illumina NovaSeq | SAMN45077174 | SRR35923991 SRR35923992 |
| UAE_SA_Phar_09 | NEBNext Ultra II Directional mRNA | Illumina NovaSeq | SAMN45077175 | SRR35924402 SRR35924403 |
| UAE_SA_Phar_10 | NEBNext Ultra II Directional mRNA | Illumina NovaSeq | SAMN45077176 | SRR35924393 SRR35924394 |

following adapter removal and trimming, with a mean read length of 5,071 bp (median 3,407 bp) and a read length N50 of 7,424 bp. The reads showed a mean base call quality score of 14.9, and the longest read sequenced was 991,055 bp with a quality score of 10.

The estimated genome size using a k-mer approach ($k$ = 21) was 599 Mb (Figure 2). The analysis revealed a diploid genome ($p$ = 2) that is predominantly homozygous (98% AA) with very little heterozygosity (1.96% AB). While GenomeScope profiles are not optimized for standard ONT reads (given the lower accuracy in comparison to short reads), the use of ONT-derived k-mer counts still provides a valuable and informative approximation of genome size, particularly in the absence of complementary short-read sequencing. The genome size estimated here is consistent with that of other assembled *Porites* genomes [35].

The initial draft assembly with very high completeness (see 'Data validation') was decontaminated (i.e., contigs mapping to non-cnidarian species were removed; Figure 3) and polished, and contigs shorter than 200 bp were removed to improve accuracy and eliminate spurious fragments. Additionally, the mitogenome of *P. harrisoni* was assembled as a single contig using Canu [57] and circularized using Circlator v1.5.5 [58], resulting in a total length of 18,639 bp closely matching a published *P. harrisoni* mitogenome from the southern Red Sea [54].

The final genome assembly (nuclear genome and mitogenome) covered 1,883 contigs with a total length of 626.7 Mb and a contig N50 of 807.4 kb (Figure 4; Table 2). Repetitive

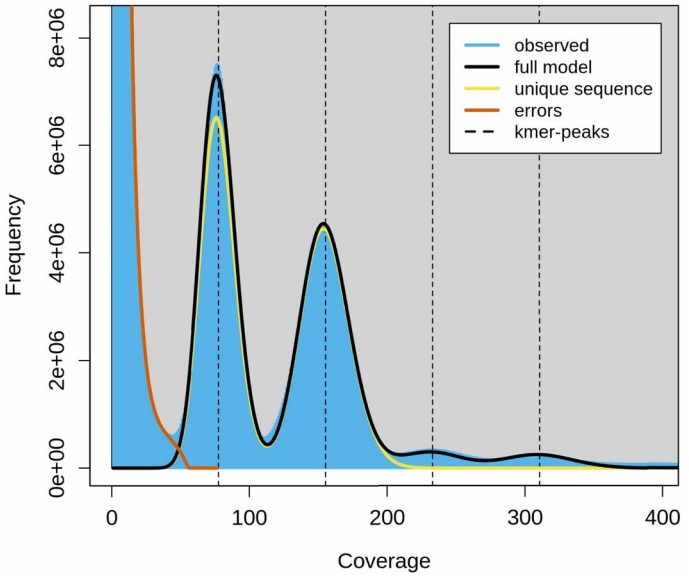

**Figure 2.  Genome-size estimation-plot of the *Porites harrisoni* genome.**
The genome-size estimation-plot was generated from a k-mer distribution ($k = 21$) using Meryl v1.3 and GenomeScope 2.0, based on ONT reads with a low error rate (1.02%). The analysis revealed a diploid genome ($p = 2$) with a size of 599 Mb, largely unique (50.3%) and predominantly homozygous (98% AA), with very little heterozygosity (1.96% AB).

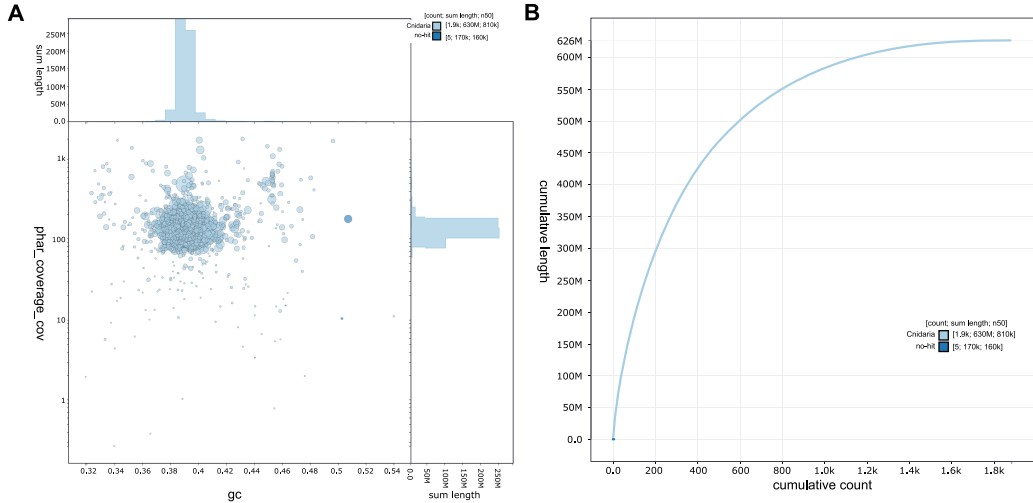

**Figure 3.  Quality assessment of the *Porites harrisoni* genome assembly using BlobToolKit v4.3.0.**
(A) GC-coverage plot showing sequence coverage (vertical axis) and GC content (horizontal axis) following decontamination (i.e., removal of contigs mapping to non-cnidarian species). Contigs are colored by phylum (light blue = Cnidaria, dark blue = no hit); circles are sized in proportion to contig length. Histograms show the distribution of contig length sums along each axis. (B) Cumulative sequence length of the final assembly after removal of contigs from non-target species. Curves show cumulative length for all contigs with colors denoting phylum (light blue = Cnidaria, dark blue = no hit).

sequences identified through a combination of RepeatModeler and EDTA [61, 62] comprised 371,184,681 bp (59.23% of the assembled genome), including retroelements (15.89%), DNA

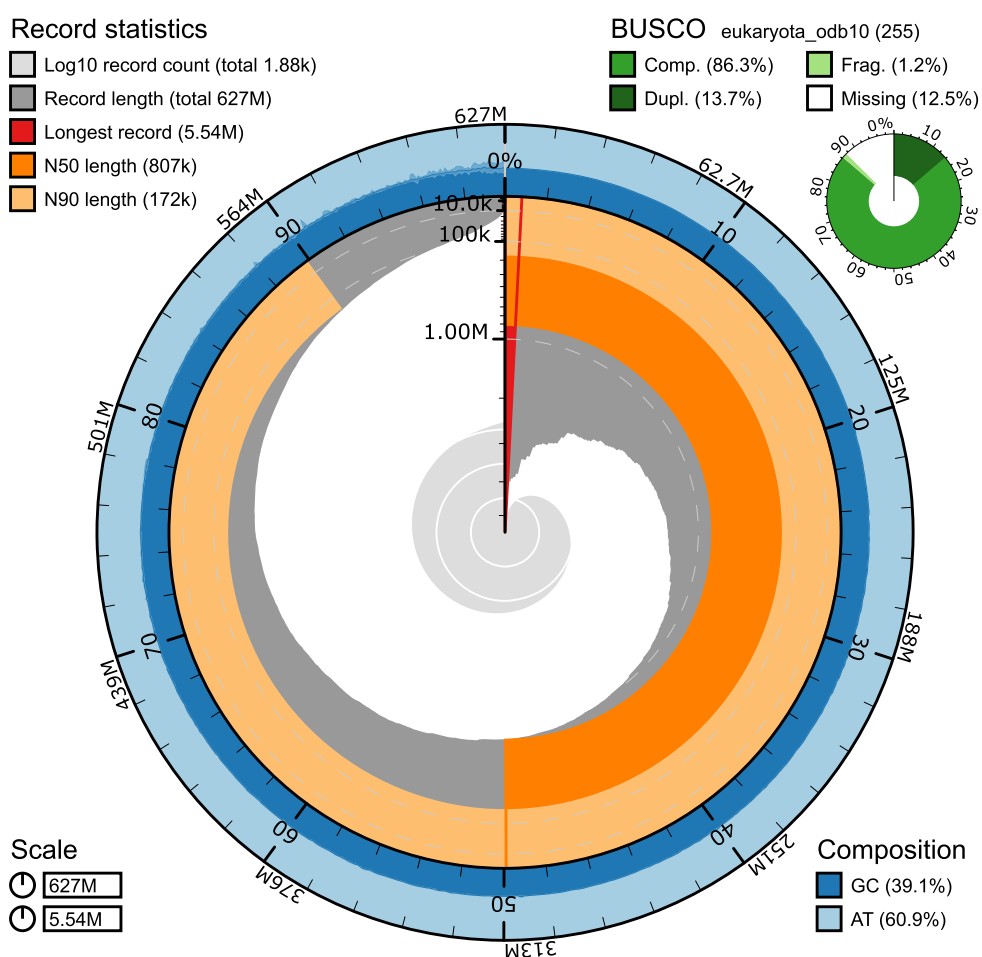

**Figure 4.** **Assembly of the *Porites harrisoni* genome.**
The snail plot generated by BlobToolKit v4.3.0 is divided into 1,000 size-ordered bins around the circumference, with each bin representing 0.1% of the 626,672,702 bp assembly. The distribution of sequence lengths is shown in dark grey with the plot radius scaled to the longest sequence present in the assembly (5,540,956 bp, shown in red). Orange and pale-orange arcs show the N50 (807,480 bp) and N90 (172,108 bp) sequence lengths, respectively. The pale grey spiral shows the cumulative sequence count on a log scale, with white scale lines showing successive orders of magnitude. The blue and pale-blue area around the outside of the plot shows the distribution of GC, AT, and N percentages in the same bins as the inner plot. A summary of complete, duplicated, fragmented, and missing BUSCO genes in the eukaryota_odb10 set is shown at the top right (Table 2).

transposons (10.00%), and unclassified repeats (31.71%; Table 3). This number of repetitive sequences is higher than for other *Porites* genomes, which have between 40 to 50% repeat content [35], and is likely due to the combined use of multiple repeat-identification methods rather than a single approach.

## Genome annotation

The *Porites harrisoni* genome assembly was annotated using the BRAKER3 pipeline [64, 89], and functional annotation was done using funannotate [52]. The resulting annotation included 27,823 protein-coding genes and 674 high-confidence eukaryotic tRNAs. The average gene length was 10,193 bp, and the average protein length was 479 amino acids (aa; Table 4). The number of protein-coding genes is lower than in other *Porites* genomes

**Table 2.** Genome assembly statistics of *Porites harrisoni.*

| General statistics | |
|---|---|
| Estimated genome size (Mb) | 599 |
| Assembled genome size (Mb) | 626.7 |
| Number of contigs | 1,883 |
| Contig N50 (kb) | 807.4 |
| Longest contig (Mb) | 5.5 |
| Shortest contig (bp) | 487 |
| Average contig length (kb) | 332.8 |
| Repeat content (%) | 59.23 |
| GC content (%) | 39.06 |
| Number of gaps in contigs | 0 |
| **BUSCO (eukaryota_odb10, *n* = 255)** | |
| Complete (%) | 86.3 |
| - Single-copy (%) | 72.5 |
| - Duplicated (%) | 13.7 |
| Fragmented (%) | 1.2 |
| Missing (%) | 12.5 |

**Table 3.** Repetitive content of the *Porites harrisoni* genome assembly.

| Group | Repeat type | Subtype | Number of elements* | Length occupied (bp) | Percentage of sequence |
|---|---|---|---|---|---|
| *Retroelements* | | | *321,723* | *99,596,350* | *15.89* |
| | SINEs | | 20,234 | 4,048,363 | 0.65 |
| | Penelope | | 4,202 | 872,880 | 0.14 |
| | LINEs | | 181,989 | 54,437,356 | 8.69 |
| | | CRE/SLACS | 2,294 | 1,323,659 | 0.21 |
| | | L2/CR1/Rex | 94,299 | 28,218,925 | 4.50 |
| | | R2/R4/NeSL | 8,477 | 2,181,497 | 0.35 |
| | | RTE/Bov-B | 25,731 | 7,080,823 | 1.13 |
| | | L1/CIN4 | 468 | 489,642 | 0.08 |
| | LTR elements | | 115,298 | 40,237,751 | 6.42 |
| | | BEL/Pao | 6,013 | 5,910,495 | 0.94 |
| | | Ty1/Copia | 2,579 | 507,411 | 0.08 |
| | | Gypsy/DIRS1 | 23,041 | 12,224,047 | 1.95 |
| | | Retroviral | 12,561 | 2,721,185 | 0.43 |
| *DNA transposons* | | | *292,657* | *62,686,658* | *10.00* |
| | hobo-Activator | | 34,240 | 8,524,951 | 1.36 |
| | Tc1-IS630-Pogo | | 19,013 | 4,057,297 | 0.65 |
| | MULE-MuDR | | 1,125 | 211,018 | 0.03 |
| | Tourist/Harbinger | | 11,439 | 2,810,493 | 0.45 |
| *Rolling-circles* | | | *6,200* | *2,355,621* | *0.38* |
| *Simple repeats* | | | *104,971* | *7,719,084* | *1.23* |
| *Satellites* | | | *416* | *136,745* | *0.02* |
| *Unclassified* | | | *759,986* | *198,690,223* | *31.71* |
| **Total** | | | **1,485,953** | **371,184,681** | **59.23** |

Note: *repeats fragmented by insertions or deletions are counted as one element, where possible to ascertain.

(between 30,000 and 40,000 genes; [13, 35]), but is comparable to other coral genomes from the complex and robust clade [32, 90–92]. BUSCO completeness comparisons indicate a higher proportion of missing conserved genes in the *P. harrisoni* assembly (12.5% versus ~1–3% in other published *Porites* genomes), suggesting that some genes may be absent due to assembly fragmentation, limited sequencing depth, or sequencing technology differences (e.g., short-read, long-read, hybrid assembly).

**Table 4.** Genome annotation statistics for *Porites harrisoni.*

| Structural annotation | |
|---|---|
| Protein-coding genes | 27,823 |
| Average gene length (bp) | 10,193 |
| Average protein length (aa) | 479 |
| High-confidence tRNAs | 674 |
| **Proteins with available functional annotation** | |
| EggNOG | 21,927 |
| InterProScan | 21,802 |
| COG | 20,717 |
| Pfam | 19,492 |
| GO terms | 16,430 |
| SignalP | 3,959 |
| Phobius | 3,906 |
| BUSCO (metazoa_odb10, *n* = 954) | 1,211 |
| MEROPS | 1,049 |
| CAZYme | 505 |
| **BUSCO (eukaryota_odb10, *n* = 255)** | |
| Complete (%) | 83.5 |
| - Single-copy (%) | 61.2 |
| - Duplicated (%) | 22.3 |
| Fragmented (%) | 2.4 |
| Missing (%) | 14.1 |

Additionally, the mitogenome was annotated using MITOS2 [59] and manually curated in Geneious Prime (GraphPad Software, LLC, Boston, USA). The mitogenome contains 13 protein-coding genes, 2 tRNAs (trnM, trnW), and 2 rRNAs (rnL, rnS; Figure 5). Two large Group I introns interrupting the nad5 and cox1 genes, with 11,131 bp and 972 bp in length, respectively, are present. By comparison, a published *P. harrisoni* mitogenome from the southern Red Sea [54] has the same 13 protein-coding genes, the same 2 tRNA and 2 rRNA genes, and 2 similar Group I introns.

## Data validation

The initial draft assembly of the *Porites harrisoni* genome was 792 Mb in size and exhibited a high level of completeness, with a BUSCO score of 98.0% (single = 81.1%, duplicated = 16.9%, fragmented = 1.2%, missing = 0.8%) based on the eukaryota_odb10 reference set (*n* = 255), indicating that the majority of expected single-copy orthologs was successfully recovered. Following the removal of contigs not aligning to cnidarian sequences to ensure taxonomic accuracy and removal of non-coral sequences, the completeness was reassessed using BUSCO. This reassessment resulted in a completeness score of 86.3% (single = 72.5%, duplicated = 13.7%, fragmented = 1.2%, missing = 12.5%) utilizing the eukaryota_odb10 reference set (*n* = 255; Table 2). Further refinements of the *P. harrisoni* assembly could benefit from incorporating additional long-read (e.g., PacBio HiFi) and/or short-read resequencing data to improve contiguity and resolution, especially for unequivocally determining the definitive number of protein-coding genes; the integration of Hi-C data could aid in generating a chromosome-level assembly (e.g., [93]), although misjoints may be introduced if the Hi-C data is of insufficient quality. Subsequent evaluation of completeness based on the set of annotated protein-coding genes yielded a BUSCO completeness of 83.5% (single = 61.2%, duplicated = 22.3%, fragmented = 2.4%, missing = 14.1%) using the eukaryota_odb10 reference set (*n* = 255; Table 4). Although other *Porites* genomes show

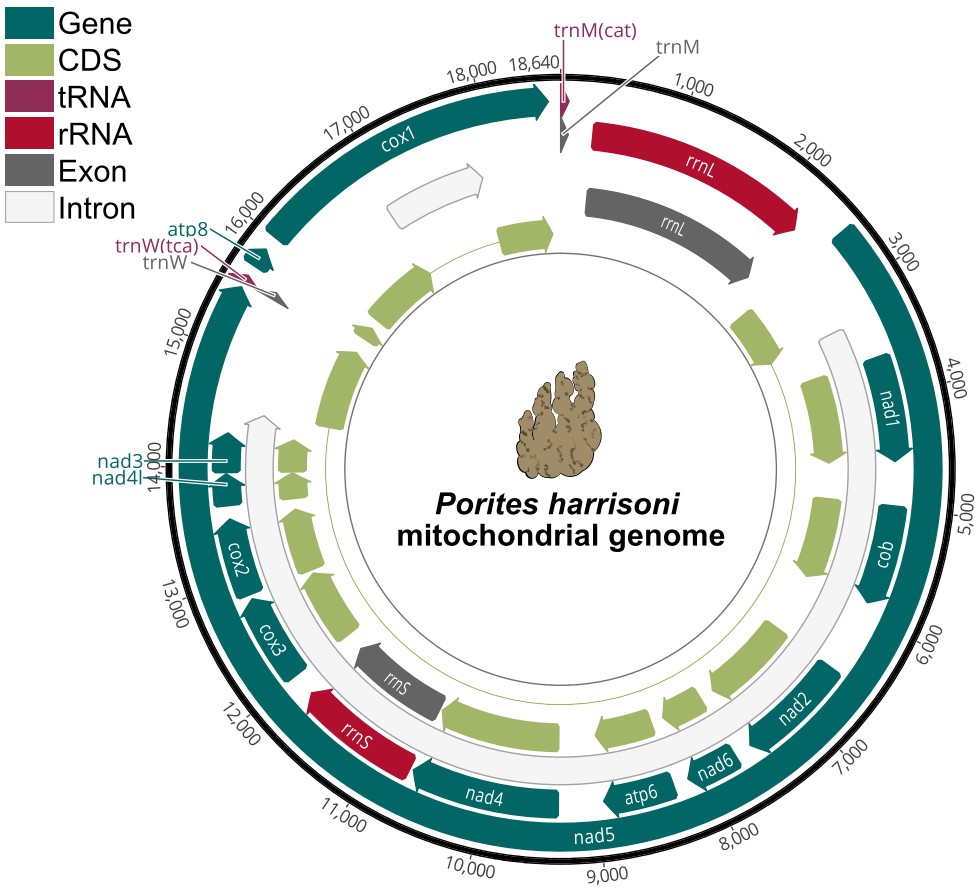

**Figure 5.** **Mitochondrial genome (mitogenome) of *Porites harrisoni*.**
The direction of the arrows indicates the read orientation of the respective genes. The annotated features are demarcated by color: teal indicates the annotated genes ($n$ = 13), light green denotes their corresponding coding sequences (CDS); tRNAs ($n$ = 2) are in dark pink, rRNAs ($n$ = 2) in red, and their respective exons are in grey ($n$ = 4); introns interrupting the nad5 and cox1 genes ($n$ = 2) are in white.

higher completeness scores [13, 35], the results in this study confirm that the final *Porites harrisoni* assembly and annotation represent a high-quality and taxonomically validated coral genome, with the reduction in BUSCO completeness attributable to the deliberate exclusion of non-cnidarian sequences, rather than a reduction in genomic accuracy.

## CONCLUSION

The genome presented in this study represents the first assembled genome of *Porites harrisoni* from the thermally extreme environment of the southern PAG. As one of the world's most thermally resilient coral species, the genome of this coral provides a critical reference for investigating the genomic basis of heat tolerance, rapid adaptation, and host-algal symbiosis under severe thermal stress. The availability of a high-quality and complete mitochondrial genome from the PAG enables comparative genomic analyses across coral species and populations, facilitating the identication of adaptive loci. Placed alongside the increasing suite of high-quality reference genomes for corals, e.g., [32, 35, 90–92, 94] and other non-model taxa [95–97], this genome strengthens the expanding

| Table 5. | Software tools: versions and sources. | |
|---|---|---|
| **Software tool** | **Version** | **Source** |
| ETE toolkit | 3.1.3 | https://github.com/etetoolkit/ete |
| MinKNOW | 23.07.12 | https://nanoporetech.com/document/experiment-companion-minknow |
| Dorado | 7.1.4 | https://github.com/nanoporetech/dorado |
| porechop | 0.2.4 | https://github.com/rrwick/Porechop |
| NanoPlot | 1.44.1 | https://github.com/wdecoster/NanoPlot |
| chopper | 0.9.0 | https://github.com/wdecoster/chopper |
| NCBI datasets | 16.14.0 | https://github.com/ncbi/datasets |
| Meryl | 1.3 | https://github.com/marbl/meryl |
| GenomeScope2.0 | 2.0 | https://github.com/tbenavi1/genomescope2.0 |
| NECAT | 0.0.1 | https://github.com/xiaochuanle/NECAT |
| BlobToolKit | 4.3.0 | https://github.com/genomehubs/blobtoolkit |
| Racon | 1.5.0 | https://github.com/lbcb-sci/racon |
| Medaka | 1.12.0 | https://github.com/nanoporetech/medaka |
| funannotate | 1.8.5 | https://github.com/nextgenusfs/funannotate |
| BUSCO | 6.0.0 | https://busco.ezlab.org/busco_userguide.html |
| minimap2 | 2.28 | https://github.com/lh3/minimap2 |
| Canu | 2.3 | https://github.com/marbl/canu |
| Circlator | 1.5.5 | https://github.com/sanger-pathogens/circlator |
| MITOS2 | 2.1.9 | https://gitlab.com/Bernt/MITOS |
| Geneious Prime | 2025.1.1 | https://www.geneious.com/ |
| RepeatModeler | 2.0.4 | https://github.com/Dfam-consortium/RepeatModeler |
| EDTA | 2.2.2 | https://github.com/oushujun/EDTA |
| RepeatMasker | 4.1.6 | https://github.com/Dfam-consortium/RepeatMasker |
| FastQC | 0.12.1 | https://github.com/s-andrews/FastQC |
| Trimmomatic | 0.39 | https://github.com/usadellab/Trimmomatic |
| BRAKER3 | 3.0.8 | https://github.com/Gaius-Augustus/BRAKER |
| STAR | 2.7.11b | https://github.com/alexdobin/STAR |
| SAMtools | 1.16.1 | https://github.com/samtools/samtools |
| tRNAscan-SE | 2.0.12 | https://github.com/UCSC-LoweLab/tRNAscan-SE |
| AGAT | 1.4.1 | https://agat.readthedocs.io/en/latest/index.html |
| GenomeTools | 1.6.5 | https://github.com/genometools/genometools |
| GffRead | 0.12.7 | https://github.com/gpertea/gffread |
| InterProScan | 5.59–91.0 | https://github.com/ebi-pf-team/interproscan |
| EggNOG-mapper | 2.1.12 | https://github.com/eggnogdb/eggnog-mapper |
| Phobius | 1.01 | https://phobius.sbc.su.se/ |
| GeneForge | 1.0 | https://github.com/SequAna-Ukon/GeneForge |

genomic foundation that is driving advances in evolutionary biology and conservation science. Moreover, this genome serves as a foundation for future studies on coral evolutionary genomics and population structure of *P. harrisoni* and will support studies aimed at predicting coral responses to ongoing ocean warming.

## AVAILABILITY OF SOURCE CODE

All bioinformatic tools used in this study are listed in Table 5; all curated pipelines are available on GitHub at https://github.com/SequAna-Ukon/Porites_harrisoni_genome (MIT license). The pipelines are implemented in Bash and run on Unix-like operating systems.

## DATA AVAILABILITY

Raw gDNA sequencing data are deposited at NCBI under the BioProject PRJNA1111311 and raw RNA-Seq data are deposited under the BioProject PRJNA1354406 , both accessible under the Umbrella BioProject PRJNA749006. This Whole Genome Shotgun project has been deposited at DDBJ/ENA/GenBank under the accession JBDLLT000000000. The version

described in this paper is version JBDLLT020000000. Assembly and annotation files can be downloaded via https://www.ncbi.nlm.nih.gov/datasets/genome/GCA_040938025.2/.

## ABBREVIATIONS

aa: amino acids; CBASS: Coral Bleaching Automated Stress System; CDS: coding sequences; GO: Gulf of Oman; LINEs: long interspersed nuclear elements; LTRs: long terminal repeats; NCCT: NGS Competence Centre Tübingen; ONT: Oxford Nanopore Technologies; PAG: Persian/Arabian Gulf; SINEs: short interspersed nuclear elements; UTR: untranslated region.

## DECLARATIONS

### Ethics approval and consent for publication

Not applicable.

### Competing interests

The authors declare that they have no competing interests.

### Authors' contributions

CRV conceived and conceptualized the study. AF, GP, HM, RA, and CRV collected the samples. JAB provided field support. AF and RA processed the samples. AF and AS assembled and annotated the genome. AF visualized the results. AF, AS, and CRV interpreted all analyses and results. AF, AS, RA, and CRV wrote the manuscript draft. All authors edited and approved the final manuscript.

### Funding

AF and CRV were supported by the Deutsche Forschungsgemeinschaft (DFG, German Research Foundation) under project number 458901010. AS was funded by the Sequencing Analysis (SequAna) Core Facility at the Department of Biology at the University of Konstanz. RA and CRV were supported by the Paul G. Allen Family Foundation (PGAFF) project 'Global Search for Genetic Regulators of Coral Resilience to Thermal Stress' (G-12948). JAB was supported by funding from Tamkeen under the NYUAD Research Institute grant CG009 to Mubadala ACCESS and grant CGSB5 to CGSB.

### Acknowledgements

We thank the NYU Abu Dhabi Core Technology Marine Science group and particularly Dain McParland for field support, and we thank Luigi Colin for technical support. We gratefully acknowledge the Environment Agency Abu Dhabi research permit with the reference number 56930. We thank the NGS Competence Center Tübingen (NCCT) for their sequencing services and technical assistance (project number QCVTS-SEQ1077).

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
