## [Reviewer Report]

Reviewer name and names of any other individual's who aided in reviewer Oleg SimakovDo you understand and agree to our policy of having open and named reviews, and having your review included with the published papers. (If no, please inform the editor that you cannot review this manuscript.)YesIs the language of sufficient quality?YesPlease add additional comments on language quality to clarify if needed
Are all data available and do they match the descriptions in the paper? YesAdditional CommentsAre the data and metadata consistent with relevant minimum information or reporting standards? See GigaDB checklists for examples <a href="http://gigadb.org/site/guide" target="_blank">http://gigadb.org/site/guide</a>YesAdditional CommentsIs the data acquisition clear, complete and methodologically sound?YesAdditional CommentsIs there sufficient detail in the methods and data-processing steps to allow reproduction?YesAdditional CommentsIs there sufficient data validation and statistical analyses of data quality? YesAdditional CommentsIs the validation suitable for this type of data?YesAdditional CommentsIs there sufficient information for others to reuse this dataset or integrate it with other data?YesAdditional CommentsAny Additional Overall Comments to the AuthorOverall, a very useful resource, the manuscript is clearly written and the data is consistently described. The genome assembly and annotation is well executed given the available data. Two minor suggestions: - include a sentence or two on potential genome assembly improvements (and if any pitfalls can be encountered), for example using HiC data and/or long-read (re)sequencing. - specify explicitly which "shorter reads" (Nanopore?) were used for ONT polishing in the assembly section and their amount.RecommendationAccept

---

## [Reviewer Report]

Reviewer name and names of any other individual's who aided in reviewer Yue SongDo you understand and agree to our policy of having open and named reviews, and having your review included with the published papers. (If no, please inform the editor that you cannot review this manuscript.)YesIs the language of sufficient quality?YesPlease add additional comments on language quality to clarify if needed
Are all data available and do they match the descriptions in the paper? YesAdditional CommentsAre the data and metadata consistent with relevant minimum information or reporting standards? See GigaDB checklists for examples <a href="http://gigadb.org/site/guide" target="_blank">http://gigadb.org/site/guide</a>YesAdditional CommentsIs the data acquisition clear, complete and methodologically sound?YesAdditional CommentsIs there sufficient detail in the methods and data-processing steps to allow reproduction?YesAdditional CommentsThe authors provide a standard and well-documented methodology: assembly, transposable-element annotation, and gene structural annotation all rely on widely used software and established pipelines. The parameter settings and post-assembly processing steps have been described.Is there sufficient data validation and statistical analyses of data quality? NoAdditional CommentsWith respect to data quality, the authors note that the number of annotated protein-coding genes in Porites harrisoni is lower than that reported for other congeneric species, yet they offer no further discussion. This discrepancy is striking and warrants clarification: is it a biological reality reflecting gene loss or genome compaction in this species, or is it an artefact arising from differences in annotation pipelines, gene-model thresholds, or assembly completeness among studies? A concise comparative analysis—and explicit acknowledgment of methodological variables—would help readers properly interpret this genomic feature.Is the validation suitable for this type of data?YesAdditional CommentsIs there sufficient information for others to reuse this dataset or integrate it with other data?NoAdditional CommentsAlthough the authors present a valuable and rare coral genome assembly, the manuscript appears to offer only basic genomic data. There is limited elaboration on the declared aim of illuminating the molecular basis of thermal tolerance. In particular, after the structural annotation of protein-coding genes, no systematic functional characterization (e.g., GO/KEGG enrichment, comparative analyses of heat-stress-related gene families, or symbiosis-related pathways) is provided. This section seems to have been undertaken but is neither described nor discussed in the current version.Any Additional Overall Comments to the Author(1) The quality of the figures could be further improved. Specifically, in Figure 1, the phylogenetic tree appears to be hand-drawn and lacks the polish typically seen in published phylogenetic analyses. It is recommended that the authors refer to examples from other studies for guidance on improving visual quality. Additionally, the tree currently lacks common indicators of phylogenetic robustness, such as bootstrap values or other support metrics. (2) In panel A (Figure 1), species highlighted in brown are presumably those included in this study. It would be helpful to add a legend clarifying the meaning of the different font colors to improve readability. Furthermore, the labeling format for sub-figures is inconsistent across the manuscript—for example, “Figure 1A, B” in one instance and “Figure 2A, B” in another. Standardizing the labeling format throughout would enhance clarity and professionalism. (3) Line 273: There appears to be an error in the unit used for “average protein length.” If this value refers to the length of the encoded proteins, it should not be expressed in base pairs (bp). Please clarify the meaning and use the appropriate unit (e.g., amino acids).RecommendationMinor Revision